# Comparison between Conventional Blind Injections and Ultrasound-Guided Injections of Botulinum Toxin Type A into the Masseter: A Clinical Trial

**DOI:** 10.3390/toxins12090588

**Published:** 2020-09-11

**Authors:** Hyungkyu Bae, Jisoo Kim, Kyle K. Seo, Kyung-Seok Hu, Seong-Taek Kim, Hee-Jin Kim

**Affiliations:** 1Division in Anatomy and Developmental Biology, Department of Oral Biology, Human Identification Research Institute, BK21 PLUS Project, Yonsei University College of Dentistry, 50-1 Yonsei-ro, Seodaemun-gu, Seoul 03722, Korea; hkbae410@yuhs.ac (H.B.); hks318@yuhs.ac (K.-S.H.); 2Youth Clinic, 30 Apgujeong-ro 80-gil, Gangnam-gu, Seoul 03722, Korea; topandbest1@gmail.com; 3Modelo Clinic, 21 Apgujeong-ro 60-gil, Gangnam-gu, Seoul 03722, Korea; doctorseo@hotmail.com; 4Department of Orofacial Pain and Oral Medicine, Yonsei University College of Dentistry, 50-1 Yonsei-ro, Seodaemun-gu, Seoul 03722, Korea; 5Department of Materials Science & Engineering, College of Engineering, Yonsei University, 50 Yonsei-ro, Seodaemun-gu, Seoul 03722, Korea

**Keywords:** paradoxical masseteric bulging, ultrasonography guided injection, botulinum neurotoxin type A injection, compensatory hypertrophy

## Abstract

The aim of the study was to propose a more efficient and safer botulinum toxin type A (BoNT-A) injection method for the masseter by comparing the conventional blind injection and a novel ultrasonography (US)-guided injection technique in a clinical trial. The 40 masseters from 20 healthy young Korean volunteers (10 males and 10 females with a mean age of 25.6 years) were included in this prospective clinical trial. The BoNT-A (24 U) was injected into the masseter of each volunteer using the conventional blind and US-guided injection techniques on the left and right sides, respectively, and analyzed by US and three-dimensional (3D) facial scanning. One case of PMB (paradoxical masseteric bulging) was observed on the side where a conventional blind injection was performed, which disappeared after the compensational injection. The reduction in the thickness of the masseter in the resting state differed significantly at 1 month after the injection between the conventional blind injection group and the US-guided injection group by 12.38 ± 7.59% and 17.98 ± 9.65%, respectively (*t*(19) = 3.059, *p* = 0.007). The reduction in the facial contour also differed significantly at 1 month after the injection between the conventional blind injection group and the US-guided injection group by 1.95 ± 0.74 mm and 2.22 ± 0.84 mm, respectively (*t*(19) = 2.908, *p* = 0.009). The results of the study showed that the US-guided injection method that considers the deep inferior tendon by visualizing the masseter can prevent the PMB that can occur during a blind injection, and is also more effective.

## 1. Introduction

Botulinum toxin type A (BoNT-A) was first used for weakening the eye muscles, and its application was expanded by Moore and Wood in 1994 as a treatment for masseter hypertrophy [1,2,3]. The demand for BoNT-A injections into the masseter muscle is increasing markedly, especially among some Asian populations in which it is assumed that the ovoid facial shape and slim lower face is optimal [3,4,5]. These injections are used not only for aesthetic purposes to reduce masseter hypertrophy, but also for therapeutic purposes to reduce bruxism and clenching [4,6]. The effects of BoNT-A injections are reversible and safe, with no significant side effects when the appropriate dose is administered.

Paradoxical masseteric bulging (PMB) is one of the most frequent sequelae when injecting BoNT-A into the masseter. Lee et al. (2017) reported the occurrence of PMB after injecting BoNT-A into the masseter, while Nam-Ho et al. (2005) reported that uneven bulging of muscles on the facial surface occurred 2 to 4 weeks after injections, especially in male patients with thin facial skin [7,8]. The incidence of PMB varies greatly depending on the literature and is reported from 0.15% to 27.3% [9]. Currently, the most common approach is to inject BoNT-A deeply into the lower third of the masseter, in order to avoid diffusion or unintended injection into surrounding tissues, including the facial expression muscles and parotid gland [10]. Lee et al. (2017) suggested that the deep inferior tendon (DIT) located inside the muscle may cause PMB by preventing the spread of toxin when the deep injection method is used [7]. However, it is nearly impossible to determine the internal location of the DIT with the naked eye or by palpating, which represents a limitation of blind injections based on the presence of the DIT.

Ultrasonography (US) has recently been utilized in guided injections as well as examinations in musculoskeletal areas [11,12,13,14]. Since the target muscle and needle position can be identified in real time, US is used when targeting small muscles or precisely injecting them at a specific position. The ability to visualize anatomical variations provides safety and reliability during US-guided injections.

The aim of the study was to propose a more efficient and safer BoNT-A injection method for the masseter by comparing the conventional blind injection and a novel US-guided injection technique in a clinical trial.

## 2. Results

### 2.1. A Case of Paradoxical Masseteric Bulging

One case of PMB was observed on the side where a conventional blind injection was performed. Soft bulging was evident in the anteroinferior area of the masseter while clenching at 3–4 days after the procedure (Figure 1). An immediate injection of 8 U of BoNT-A was performed to reduce the PMB. The superficial part of the masseter showed localized excessive contraction in the US image, which disappeared after the compensational injection, along with reduction of the PMB (Figure 2). There were no other side effects or complaints.

### 2.2. US-Based Analysis of the Thickness of the Masseter

The thicknesses of the masseters on the left and right sides were 12.33 ± 2.92 mm (mean ± SD) and 13.00 ± 3.26 mm, respectively, before the injection, and 10.74 ± 2.77 mm and 10.81 ± 2.91 mm 1 month after the injection. The reduction in the thickness of the masseter in the resting state differed significantly at 1 month after the injection between the conventional blind injection group and the US-guided injection group (Figure 3) by 12.38 ± 7.59% and 17.98 ± 9.65%, respectively (*t*(19) = 3.059, *p* = 0.007) (Table 1).

### 2.3. Three-Dimensional Analysis of the Facial Contour

Most (85%, *n* = 17) of the 20 volunteers showed a greater reduction in the facial contour on the US-guided injection side than on the blind injection side. The reduction in the facial contour was 1.95 ± 0.74 mm in the conventional blind injection group and 2.22 ± 0.84 mm in the US-guided injection group (*t*(19) = 2.908, *p* = 0.009) (Table 2).

## 3. Discussion

Several decades have passed since the BoNT-A injection was first applied to the masseter for cosmetic and therapeutic purposes. The conventional guideline is to inject deeply into three or four locations on the lower third of the masseter, and this is still commonly used [10]. This is a method of deep injection in which the needle tip contacts the bony surface of the mandible, and it minimizes the adverse effects on the surrounding anatomical structures. The origin of the risorius muscle generally covers the anterior third of the masseter on the superficial musculoaponeurotic system layer. Therefore, injecting the toxin too anteriorly or shallowly can result in unintended paralysis of the risorius muscles, which can shorten the mouth or result in facial asymmetry when grinning [15]. Deep injections to the bony surface also help clinicians prevent unintentionally injecting into the parotid gland, which usually covers the posterior one-fourth of the masseter [16]. Furthermore, the masseteric nerve is known to innervate the lower part of the muscle, and so injecting into the lower third of the masseter would maximize the effect of the toxin [17].

While BoNT-A injections are known to be relatively safe when the appropriate dose is used, there are still some adverse effects such as PMB that are of concern to both clinicians and patients when using BoNT-A to treat the masseter. The etiology of PMB is still unclear, but it has been assumed that the deep part of the muscle is less affected when the toxin is injected deeply [18]. It is known that individual differences in the contractions of the three layers of the masseter result in various types of bulging [19]. It has also been reported that there is a strong tendinous DIT inside the masseter that may prevent the spread of toxin, as well as individual differences in the morphology of the DIT [7,20]. It can therefore be inferred that imbalance of the contractile capability of the masseter induced by the deep injection method and the diffusion-inhibiting effect of DIT can cause PMB due to excessive contraction of the superficial layer. Although BoNT-A is known to spread a few centimeters from the needle tip immediately after it is injected, a thick muscle and diffusion restriction by the DIT can prevent the toxin from spreading across the muscle when using that conventional blind method in which the injection is performed just above the bone surface [21].

In order to minimize the probability of PMB occurrence, it is recommended to inject through the entire masseter while considering the DIT. A method of pulling back the needle after contacting the bony surface has been suggested to prevent PMB [7]. However, in the actual blind injection procedure, clinicians cannot locate the location of the DIT due to its morphological variations between individuals. In addition, it has been reported that when a blind injection is performed in which the skin is premarked and injected percutaneously, the toxin is frequently injected too anteriorly (up to 40% of cases), or the needle does not reach the desired depth due to it being too short (up to 20% of cases) [22].

A US-guided injection can be utilized instead of a blind injection to accurately locate the DIT and perform a precise injection. The real-time visualization of intramuscular structures and needle location allows targets to be accurately located, and this also helps the clinician to avoid structures that should not be affected, including facial expression muscles, blood vessels, and the parotid gland. US-guided injections have already been used in the detection and targeting of small structures such as the risorius muscle [23]. It is expected that if US-guided injections are used to selectively inject into the multilayer masseter, the muscle reduction will be more effective, and PMB will be prevented.

The present study analyzed changes in muscle thickness and overall facial contour at 1 month after US-guided and conventional blind injections, which revealed a significant difference in the reduction of the masseter (17.98 ± 7.59% and 12.38 ± 9.65%, respectively; *p* = 0.007) and in the facial contour change (−2.22 ± 0.84 mm and −1.95 ± 0.74 mm; *p* = 0.009). These results suggest that, in contrast with the conventional method, a US-guided injection distributes the toxin across all regions, resulting in a volume reduction effect over the entire area. Reducing both the deep and superficial parts of the superficial layer in consideration of the DIT through a US-guided injection is not only meaningful for preventing PMB, but also influences the amount of volume reduction and facial contour changes. Regarding adverse effects, one case of PMB occurred in the conventional blind injections. This number is too small to statistically confirm that US-guided injection is effective in reducing the incidence of PMB, and so more cases need to be evaluated in future studies.

US-guided injections of the masseter have several advantages over conventional blind injections. First, this method allows the injection to be performed while considering the location of the DIT. Clinicians can prevent PMB by injecting into parts of the muscle that are both deeper and more superficial than the DIT. Furthermore, if necessary, a small-dose injection can be applied only at the desired location depending on the bulging type. Second, US-guided injection makes it possible to identify the location of the masseter and surrounding structures in real-time before and during the procedure. The locations of the boundary of the masseter, the parotid gland on the posterior side of the masseter, and the risorius and buccinator muscles on the anterior side of the masseter can be determined in order to prevent affecting these structures as well as ensuring that the toxin is injected outside the muscle. Third, a larger volume reduction effect can be obtained with the same dose of toxin. In general, it is known that the toxin dose is not associated with the amount of volume reduction above a certain dose threshold. However, the results of the present study show that greater muscle thickness and facial contour reductions can be obtained through a US-guided injection.

A US-based evaluation provides many clues about the anatomy of the masseter and the clinical application of BoNT-A. In addition, this study has demonstrated that a US-guided injection method that considers the DIT by visualizing the masseter can prevent the PMB that can occur during a blind injection, and is also more effective.

## 4. Materials and Methods

### 4.1. Subjects

The 40 masseters from 20 healthy young Korean volunteers (10 males and 10 females with a mean age of 25.6 years) were included in this prospective clinical trial. All of the volunteers had a complaint of a bulky masseter and 4 of them had a myofascial masseteric pain. Before participating in the study, a signed written-consent form was obtained from each volunteer. Each volunteer was a dental student or staff member at the College of Dentistry, Yonsei University, Seoul, Korea. Potential side effects were fully explained to the subjects, and they were free to withdraw from the treatment and research at any time. The exclusion criteria for this study included pregnancy, a history of drug allergy, any other serious medical illnesses, or surgical or nonsurgical treatment in the facial area (including BoNT-A injection) within the previous 6 months. All of the study procedures were approved by the institutional review board of the Yonsei University College of Dentistry (IRB No. 2-2019-0008), approval date: 11 April 2019.

### 4.2. Reference Lines

The following reference lines were designated for the BoNT-A injections, US measurements, and 3D scanning (Figure 4): (1) the line passing through the cheilion and otobasion inferius (designated as T1); (2) the line corresponding to the lower margin of the mandible (designated as T3); (3) the line located between T1 and T3 (designated as T2); and (4) the lines trisecting the masseter longitudinally (designated as L1 and L2). All of these reference lines were drawn on the skin with waterproof eyeliner prior to performing the injections and measurements.

### 4.3. Injection Method

Using the conventional blind and US-guided injection techniques on the left and right sides, respectively, 24U of BoNT-A (Innotox, 1.25 mL/vial, 4 units/0.1 mL; Medytox Inc., Seoul, South Korea) was injected into the masseter of each volunteer. The injection locations for the masseter were determined in accordance with the conventional blind injection guidelines. BoNT-A was injected at four crossing points of the reference lines (T1 and L1, T1 and L2, T2 and L1, and T2 and L2) for each masseter. To avoid producing an iatrogenic tattoo with the marking ink, the actual injection was performed adjacent to the indicated injection location. The site of maximal protrusion during clenching was included among the four injection locations.

On the left side of each volunteer, a conventional injection was performed with an 8-mm-long 30-G BD syringe, which is widely used in clinical practice. The needle was inserted perpendicularly at each injection location. At each of the four locations, the needle tip was inserted until it pressed lightly against the bone, and then 6 U of BoNT-A was injected (Figure 5A).

On the right side of each volunteer, a US-guided injection was performed with a 25-mm-long 30-G needle to cover the whole masseter without applying any pressure. After confirming the locations of the masseter, internal DIT, and surrounding structures using a US device, a US-guided injection was performed using an out-of-plane technique with the transducer located along T1 and T2 (Figure 5B). The injection locations were the same as in the conventional blind injection, but BoNT-A was injected into parts that were both deeper and more superficial than the DIT (Figure 6). Moreover, not only perpendicular movements of the needle tip along each injection location, but also backward and forward movements, were performed to achieve precise injections according to the shape and DIT type of the masseter under US guidance. The dose of the toxin was divided based on the thicknesses of the superficial and deep parts. The total dose (24 U) of the US-guided injections was the same as that of the conventional blind injections. All of the injections were performed by the same person. Each volunteer was examined using three-dimensional (3D) facial scanning and US imaging before and after the injection. Volunteers were guided to contact immediately if there were any side effects or discomfort, such as PMB. The status of volunteers was examined one month after the injection, when they revisited for a US imaging and a facial scanning.

### 4.4. US-based Analysis of the Thickness of the Masseter

The patients were placed in a semisupine position for the US examination and injection. US images of the masseters on 40 sides were obtained using a real-time two-dimensional B-mode US device with a high-frequency (15 MHz) linear transducer (E-CUBE 15 Platinum, ALPINION Medical Systems, Seoul, Korea). 

The transducer was positioned just above the skin surface at each reference line, with a US gel applied thickly. Special efforts were made to avoid mechanical pressure from the US probe distorting the soft tissue. Each US video was taken before the injection and 1 month after the injection in both resting and clenching states.

A US image was taken with the probe placed horizontally on the T1 line (Figure 7). Starting in the most-relaxed state, the process of clenching and relaxing again was repeated twice, and corresponding images were taken. The thickness of the muscle was obtained by measuring the thickest part of the muscle perpendicular to the mandibular ramus. All measurements were made by one investigator who did not perform the injection. The investigator was unaware of which injection method was applied and which side the masseter was on.

### 4.5. Three-Dimensional Analysis of the Facial Contour before and after the Injection

Faces of the volunteers were scanned before and 1 month after the injection using a structured-light scanner (Morpheus3D, Morpheus Company, Seongnam, Korea). One frontal view and two oblique views were first scanned and then merged via a geometry analysis of the nearby areas at three locations (lateral canthus, alare, and cheilion) bilaterally. A 3D model was obtained using the Morpheus Plastic Solution (version 3.0) software (MPS 3.0; Morpheus Company, Seongnam, Korea).

The scanned images were superimposed using the MPS 3.0 program. The depth difference between the scanned face before and 1 month after the injection was automatically measured in the direction perpendicular to the facial surface, and is displayed in different colors in Figure 8. The location with the largest difference in each volunteer was measured and analyzed. As with the US-based analysis, all measurements were made by one investigator who did not perform the injection, and who was unaware of which injection method was applied and which side the masseter was on.

### 4.6. Statistical Analyses

The contour changes in the face measured by 3D scanning and the changes in the masseter thickness measured by US were analyzed using a paired *t*-test after the Kolmogorov–Smirnov test had been applied to check for normality. The probability criterion for statistical significance was *p* < 0.05. The statistical analysis was performed using SPSS software (version 23.0 for Windows, SPSS, Chicago, IL, USA).

## Figures and Tables

**Figure 1 toxins-12-00588-f001:**
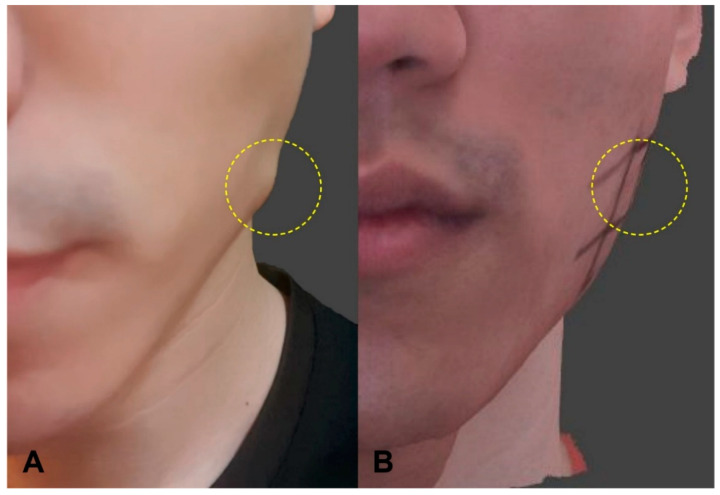
Photograph and 3D scanned models of paradoxical masseteric bulging (PMB) observed after a conventional blind injection: (**A**) Photograph showing soft bulging above the masseter while clenching; (**B**) 3D scanned model showing that the PMB was reduced after making a compensational injection into an area that was more superficial than the deep inferior tendon (DIT).

**Figure 2 toxins-12-00588-f002:**
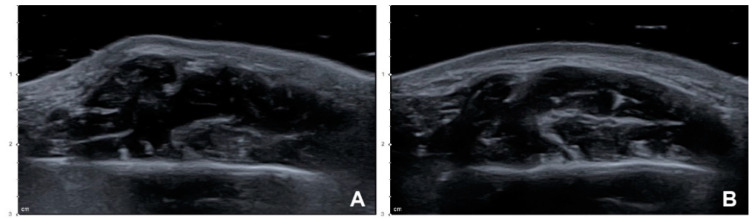
Ultrasonography (US) images of PMB after performing the conventional blind injection: (**A**) Superficial part of the masseter showing localized excessive contraction while clenching (B mode, transverse view, 15-MHz linear transducer); (**B**) localized excessive contraction in the clenching state was resolved after performing a compensational injection of 8 U of BoNT-A into the superficial part of the masseter (B mode, transverse view, 15-MHz linear transducer).

**Figure 3 toxins-12-00588-f003:**
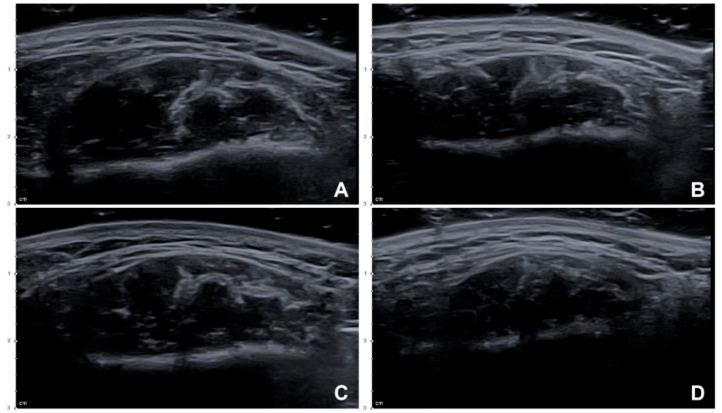
US images showing reduction of the masseter after BoNT-A injections (B mode, transverse view, 15-MHz linear transducer): (**A**) Left side, before the injection, (**B**) left side, 1 month after a conventional blind injection, (**C**) right side, before the injection, and (**D**) right side, 1 month after a US-guided injection.

**Figure 4 toxins-12-00588-f004:**
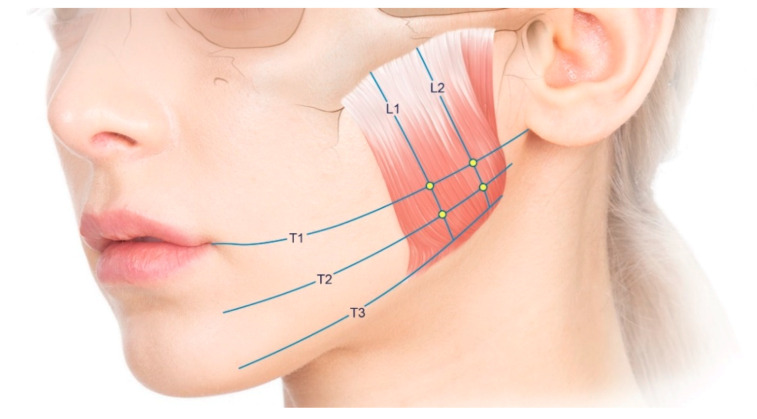
Reference lines for the injections and measurements: T1, the line passing through the cheilion and otobasion inferius; T2, the line located between T1 and T3; T3, the line corresponding to the lower margin of the mandible; L1 and L2, the lines trisecting the masseter muscle longitudinally.

**Figure 5 toxins-12-00588-f005:**
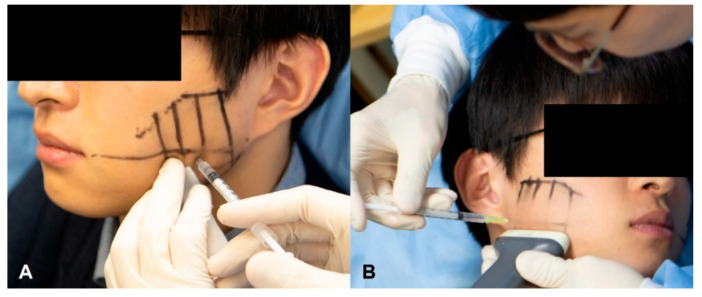
Injecting 24 U of BoNT-A (**A**) using the conventional blind injection method on the left side and (**B**) using the US-guided injection method on the right side.

**Figure 6 toxins-12-00588-f006:**
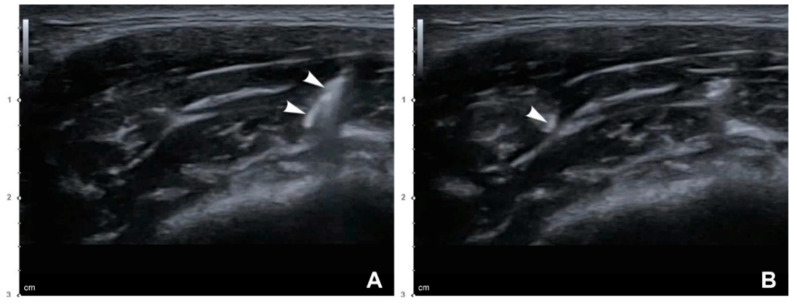
US-guided dual-plane BoNT-A injections into the masseter. White arrowheads indicate a needle tip: (**A**) deep injection (B mode, transverse view, 15-MHz linear transducer) and (**B**) superficial injection (B mode, transverse view, 15-MHz linear transducer).

**Figure 7 toxins-12-00588-f007:**
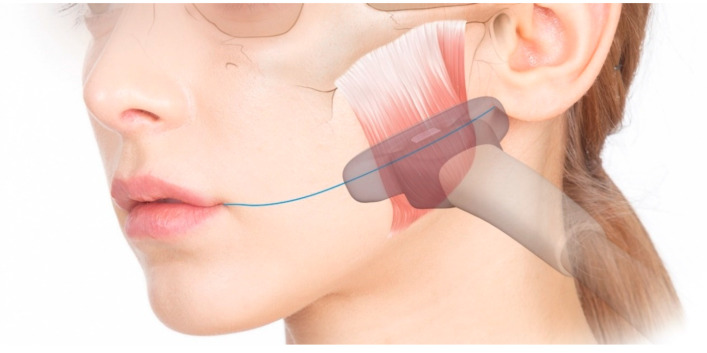
The US transducer was applied over the T1 line to obtain a US video of the masseter during the process of clenching and relaxing again, starting in the relaxed state.

**Figure 8 toxins-12-00588-f008:**
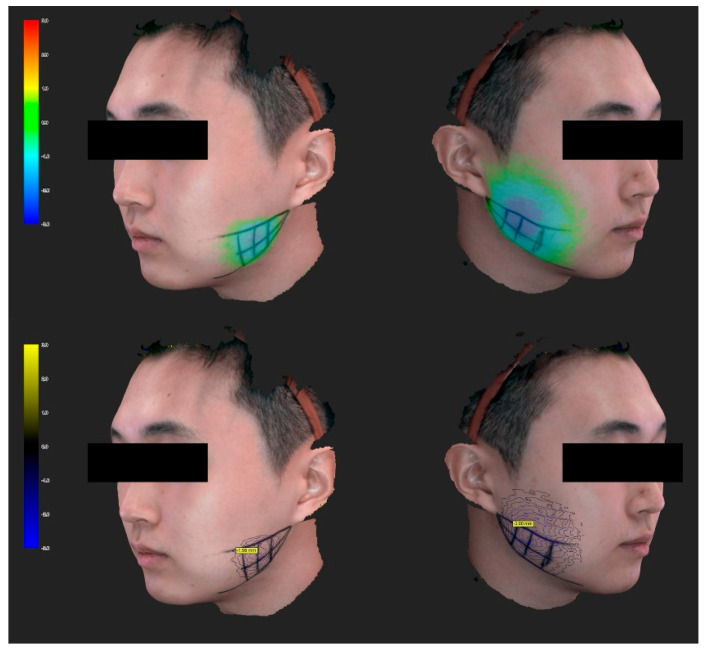
Three-dimensional scanned model of the facial changes in a volunteer. The amount of facial contour change is expressed in both colors and values. The location with the largest difference in each volunteer was measured and analyzed.

**Table 1 toxins-12-00588-t001:** Measurements of differences in the masseter muscle.

Site	Injection Method	Masseter Thickness before Injection (mm)	Masseter Thickness at 1 month after the Injection (mm)	Reduction in Masseter Thickness (%)	*p*
Left	Conventional blind injection (*n* = 20)	12.33 ± 2.92	10.74 ± 2.77	12.38 ± 7.59	0.007
Right	US-guided injection (*n* = 20)	13.00 ± 3.26	10.81 ± 2.91	17.98 ± 9.65	
Total	*n* = 40	12.66 ± 3.07	10.77 ± 2.80	15.18 ± 9.25	

Data are mean ± SD values.

**Table 2 toxins-12-00588-t002:** Measurements of the difference in the facial contour.

Site	Injection Method	Depth Difference between before and 1 month after the Injection	*p*
Left	Conventional blind injection (*n* = 20)	−1.95 ± 0.74 mm	0.009
Right	US-guided injection (*n* = 20)	−2.22 ± 0.84 mm	
Total	*n* = 40	−2.09 ± 0.80 mm	

Data are mean ± SD values.

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
