# Peer review of "Comparison between Conventional Blind Injections and Ultrasound-Guided Injections of Botulinum Toxin Type A into the Masseter: A Clinical Trial"

_toxins, 2020, doi:10.3390/toxins12090588_

Round 1

Reviewer 1 Report

This is an interesting study assessing the use of ultrasound (US) guidance of botulinum toxin injection in the masseter muscle. The goal of the present study is of interest, and the findings are novel. The clinical/technical procedures and assessments have been performed diligently by the authors. The number of study participants (n=20) is reasonable albeit not high. The authors present data suggesting a statistically significant difference of treatment effects in favor of US-guided injection compared to the conventional injection technique. There are however some issues related to the study design (or its description) that may be critical with respect of a potential bias, and that need to be addressed by the authors upon revision of the manuscript. In general, a random choice of right or left side for the conventional versus the US-guided injection might have been preferable, in order to reduce a potential bias of the subsequent assessments of treatment effects.

Major points:

  1. Methods: Were the US-guided injections and the subsequent US measurements of masseteric volumes performed by the same US investigator, or by different investigators? Please comment.
  2. Methods: If both procedures (see 1.) were performed by different investigators: Was the rater of masseteric volumes aware, or unaware, of the side of US-guided injection? Please comment.
  3. Methods: Were the facial-contour analyses performed by an independent rater? Was this rater blinded to the side of US-guided injections? Please comment.
  4. Methods/Results: Did the authors systematically screen for (and record) any side effects other than PMB? Please comment, and add this information if available.
  5. Methods/Results: Did the authors systematically record the participants’ subjective rating of improvement of complaints on the right and the left side? Please comment, and add this information if available.

Minor points:

  1. Abstract: Please explain (or avoid) the abbreviations PMB and DIT in the abstract text.
  2. Methods: Please give the exact numbers of participants with bulky masseter and of those with myofascial masseteric pain.
  3. Discussion: It would be of interest whether both, the complaints of bulky masseter and of myofascial masseteric pain, are in-label indications of botulinum toxin application in the Republic of Korea. In many countries (e.g. Germany) these applications are off-label, and therefore are not covered by health insurance. The authors may shortly comment on this.

Author Response

The authors appreciate the review's careful comments. All of the responses to the reviewer's comments are written in this note. Please see the attachment for the changed part of the manuscript (changed part is displayed in red words).

Major points:

Q1: Methods: Were the US-guided injections and the subsequent US measurements of masseteric volumes performed by the same US investigator, or by different investigators? Please comment.

A1: The authors strongly agree that this should have we should have commented on what to do to avoid potential bias and appreciate for the reviewer’s question. Masseter volume measurements were made by one investigator other than those who injected the botulinum toxin to increase the reliability. We supplemented the related description in the method part: “All measurements were made by one investigator who did not perform the injection, and the investigator was unaware of which injection method was applied and the which side the masseter is.”.

Q2: Methods: If both procedures (see 1.) were performed by different investigators: Was the rater of masseteric volumes aware, or unaware, of the side of US-guided injection? Please comment.

A2: All measurements were performed without awareness of the side or injection method. The authors agreed to mention about the investigator unawareness as commented by reviewers: “All measurements were made by one investigator who did not perform the injection, and the investigator was unaware of which injection method was applied and the which side the masseter is.”.

Q3: Methods: Were the facial-contour analyses performed by an independent rater? Was this rater blinded to the side of US-guided injections? Please comment.

A3: As answered in Q1 and Q2, the authors strongly agree that rater should be unaware of the side of US guided injection to avoid bias. In the same way as the US measurement, the investigator who did not perform the injection had performed the facial-contour analyses for this purpose, and it was conducted without information on the injection side. The authors added mention about it in the method part: “Same as the US-based analysis, all measurements were made by one investigator who did not perform the injection, and the investigator was unaware of which injection method was applied and the which side the masseter is.”.

Q4: Methods/Results: Did the authors systematically screen for (and record) any side effects other than PMB? Please comment, and add this information if available.*   

A4: The authors explained to the volunteer about the possible side effects, including PMB, before proceeding with the experiment, and guided them to contact us immediately if there was any side effects or discomfort. In addition, we checked up volunteers’ status when they visited 1 month after the injection for the US examination and facial scanning. As results, there were no complaints or additional side effects As the reviewer advised, the authors supplemented related information in methods and results: “Volunteers were guided to contact immediately if there were any side effect or discomfort such as PMB. The status of volunteers was examined one month after the injection, when they revisited for an US imaging and a facial scanning.”, “There were no other side effects or complaints.”

Q5: Methods/Results: Did the authors systematically record the participants’ subjective rating of improvement of complaints on the right and the left side? Please comment, and add this information if available.

A5: As mentioned in A4, the patient's side effects and complaints were followed up. There were no side effects or complaints other than PMB in 1 case. We did not record any subjective ratings or complaint improvements.

Minor points:

Q1: Abstract: Please explain (or avoid) the abbreviations PMB and DIT in the abstract text.

A2: The authors appreciate your advice on the abbreviations, and has modified it to make it clear.  

Q2: Methods: Please give the exact numbers of participants with bulky masseter and of those with myofascial masseteric pain.

A2: All volunteers had a bulky masseter and 4 of 20 volunteer had a myofascial masseteric pain. We have supplemented the related information in the Methods.

Q3: Discussion: It would be of interest whether both, the complaints of bulky masseter and of myofascial masseteric pain, are in-label indications of botulinum toxin application in the Republic of Korea. In many countries (e.g. Germany) these applications are off-label, and therefore are not covered by health insurance. The authors may shortly comment on this.

A3: Botulinum toxin application for bulky masseter and myofascial pain is still off-label in Korea, so it is not covered by health insurance. However, botulinum toxin treatment is recognized as the most effective and safe treatment for masseter hypertrophy, and has proven to be effective in the treatment of myofascial pain.

Reviewer 2 Report

The abreviations DIT and PMB are defined in the introduction but no in the abstracts that is the first part of the paper to be read.

Methods: What kind of BoNT-A is used in the study?, Lanzhou?, Botox?, Dysport?. It have crucial importance in the resuts. 

In fig 5 either the lines and the placement of the needle are different between both sides

Author Response

The authors appreciate the review's careful comments. All of the responses to the reviewer's comments are written in this note. Please see the attachment for the changed part of the manuscript (changed part is displayed in red words).

Q1: The abreviations DIT and PMB are defined in the introduction but no in the abstracts that is the first part of the paper to be read.

A1: The authors appreciate your advice on the abbreviations, and has modified it to make it clear.

Q2: Methods: What kind of BoNT-A is used in the study?, Lanzhou?, Botox?, Dysport?. It have crucial importance in the resuts. 

A2: Liquid-type BTX-A (Innotox, 1.25 mL/vial, 4 units/0.1 mL; Medytox Inc., Seoul, South Korea) was used in the study. We authors supplemented the related information in the Medthods: “24U of BoNT-A (Innotox, 1.25 mL/vial, 4 units/0.1 mL; Medytox Inc., Seoul, South Korea ) was injected into the masseter of each volunteer using the conventional blind and US-guided injection techniques on the left and right sides, respectively.”

Q3: In fig 5 either the lines and the placement of the needle are different between both sides

A3: In fig. 5, additional reference lines (superior, medial, lateral margin of the masseter muscle) are drawn in addition to the describe reference lines (L1, L2, T1, T2, T3). The baselines described for the injection (L1, L2, T1, T2, T3) were carefully drawn identically on both sides. In addition, as mentioned in Methods, the actual injection was performed adjacent to the indicated injection location (crossing point of reference lines). In summary, the injection point and the reference lines may look different on the figure due to the angle and additional lines, but they are same on both sides.